# Cell-in-the-loop pattern formation with optogenetically emulated cell-to-cell signaling

Melinda Liu Perkins [1✉], Dirk Benzinger[2], Murat Arcak[1] & Mustafa Khammash[2✉]

Designing and implementing synthetic biological pattern formation remains challenging due to underlying theoretical complexity as well as the difficulty of engineering multicellular networks biochemically. Here, we introduce a cell-in-the-loop approach where living cells interact through in silico signaling, establishing a new testbed to interrogate theoretical principles when internal cell dynamics are incorporated rather than modeled. We present an easy-to-use theoretical test to predict the emergence of contrasting patterns in gene expression among laterally inhibiting cells. Guided by the theory, we experimentally demonstrate spontaneous checkerboard patterning in an optogenetic setup, where cell-to-cell signaling is emulated with light inputs calculated in silico from real-time gene expression measurements. The scheme successfully produces spontaneous, persistent checkerboard patterns for systems of sixteen patches, in quantitative agreement with theoretical predictions. Our research highlights how tools from dynamical systems theory may inform our understanding of patterning, and illustrates the potential of cell-in-the-loop for engineering synthetic multicellular systems.

[1] Department of Electrical Engineering, University of California, Berkeley, CA, USA. [2] Department of Biosystems Science and Engineering, ETH Zürich, Basel, Switzerland. ✉email: mindylp@eecs.berkeley.edu; mustafa.khammash@bsse.ethz.ch

Spatial patterning is crucial for the proper functioning of diverse multicellular biological systems from slime molds[1] to developing embryos. The ability to synthetically engineer multicellular patterning will facilitate advances in designing microbial communities[2–4], creating synthetic biomaterials[5,6], and programming tissue and organ growth[7–10], among other applications[11]. While recent efforts to synthetically engineer multicellular patterning have met with success (see refs. [12–14] for reviews), relatively few of these efforts[15,16] have been guided by quantitative mathematical theory beyond numerical simulation. In contrast, conventional engineering approaches rely on the predictive power of theory both to design complex systems and to build the intuition necessary to envision new capabilities. Future progress in synthetic multicellular patterning will benefit from a firm understanding of the underlying theoretical principles, as well as scalable, efficient methods for implementing—and validating—these principles in practice.

Gene expression patterning has received much focus in the theoretical literature[17–23], and is also of particular interest in regenerative medicine, since it is central to the early stages of embryonic development and eventual cell fate determination[7,24]. There are a number of challenges associated with engineering spontaneous gene expression patterning into biochemical systems, including how to facilitate interaction among cells[25] and achieve spatial precision in the resulting patterns[26–28]. Even when successful, these implementations are still constrained by time, expense, and the availability of biological parts satisfying parameter requirements[29,30]. Moreover, it may be difficult to measure or monitor particular system components in real time, which can hinder debugging and slow down the design-build-test cycle[31].

While numerical simulation is an important method for efficient prototyping, simulations are only as valid as the models underlying them, and simplifications or faulty assumptions can limit the experimental applicability of simulation results. We propose that future efforts in synthetic patterning would benefit from an intermediate step between pure simulation and full biochemical implementation, which could be used to validate theories or incrementally test synthetic designs before they are fully incorporated into the organism. Inspired by human-in-the-loop approaches for engineering systems that must interact with complex, living individuals[32], we propose a cell-in-the-loop approach in which physical signaling among cells is substituted with computer-controlled inputs calculated in silico from real-

time measurements of gene expression. Cell-in-the-loop, by incorporating live cells into the simulation, eliminates the need to make assumptions about individual cell behavior during dynamic evolution, while retaining flexibility in testing parameters that remain under computational control. These benefits are particularly essential for patterning systems, in which the large number of interacting cells can make detailed simulations prohibitive or impossible.

We implement cell-in-the-loop using optogenetics, which have been shown to afford excellent spatiotemporal precision in applications including feedback control[33–36], and which were previously used to emulate cell-to-cell signaling for oscillatory synchronization[37]. We engineer *Saccharomyces cerevisiae* to respond to blue light[38] by increasing gene expression as measured by a fast-acting fluorescent reporter[39]. We use an optogenetic platform capable of targeting individual cells independently of each other[36], such that the light input to any given cell can be calculated based on the gene expression levels of other cells that are interacting with the target cell. Both the network architecture (which cells interact with which) as well as the exact form of interaction are programmed into the computer, allowing us to precisely modulate system parameters related to cell-to-cell signaling.

We adapt a general theory for pattern emergence in large-scale lateral inhibition systems[40,41] to inform our designs and predict steady-state outcomes. Lateral inhibition regulated by the Notch-Delta signaling pathway is responsible for patterning in a range of developmental contexts, including proneural stripe formation[42] and subsequent neural precursor selection[43] in fruit flies, as well as patterning in the central nervous system[44], inner ear[45,46], and intestine[47] of vetebrates[48]. Inspired by these systems, we program a computational signaling relation to emulate mutual inhibition among groups of cells and vary the strength of the inhibition by tuning a single digital bifurcation parameter. Once the network architecture and signaling relation are defined, inputs to cells are calculated solely based on measurements of those cells without any further external control, creating a self-contained dynamical system. Using this setup, we visualize gene expression levels of real cells by the brightness of square patches on a virtual grid (Fig. 1). We show spontaneous emergence of contrasting checkerboard patterns in which neighboring patches alternate between expressing high and low levels of gene. The theory accurately predicts which values of the bifurcation parameter produce patterns, and on average across experiments the theory also

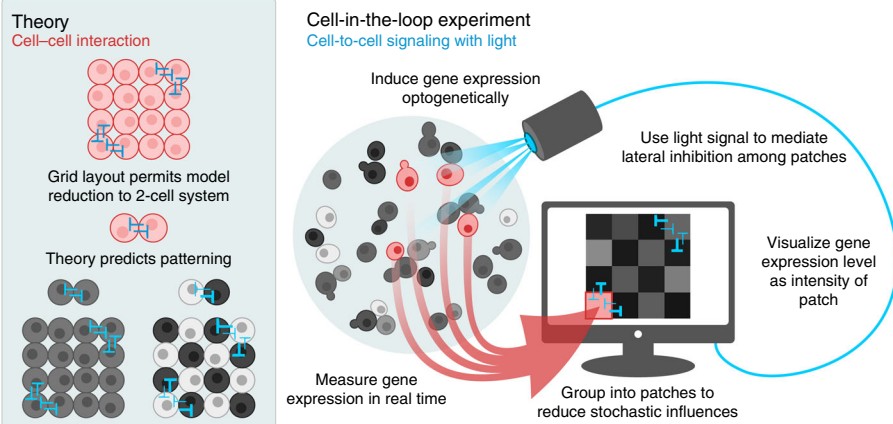

**Fig. 1 Spontaneous checkerboard patterning with optogenetically emulated cell-to-cell signaling.** Optogenetically responsive cells signal to each other through computer-controlled light inputs that vary in intensity based on the gene expression levels of other cells. We enact lateral inhibition according to the theory in Section 3.1 that predicts when cells will spontaneously separate into two classes of high and low gene expression. In all figures, red denotes in vivo and blue denotes in silico components.

quantitatively predicts contrast levels and overall patch brightness. Our results demonstrate the utility of a cell-in-the-loop approach for designing and evaluating systems of interacting cells, as well as probing the limits of deterministic theory in the face of stochastic influence.

## Results

**Theory predicts patterning using a test for bistability.** We developed theory to predict the emergence of stable contrasting patterns in deterministic systems of laterally inhibiting cells[40,41]. Here, we adapt the theory to the present optogenetic implementation. We emphasize how our system was decomposed into in vivo and in silico components, each of which corresponds to a particular element in the theory, and how this correspondence enables empirical measurement and experimental design.

Consider a system of $N$ isogenic cells signaling to each other. Suppose we measure for each cell a scalar output such as fluorescence that correlates positively with gene expression level and is designated by $w_i$ for the $i$th cell. The input $u_i$ to a cell affects output levels with an empirically characterizable dose response, which describes the steady-state level of $w_i$ for a constant-in-time input. In our setup, the input $u_i$ is light, and increasing input intensity increases gene expression. This portion of the theory represents the in vivo component of the system.

To synthesize $u_i$, we first average the measured gene expression, $w_j$, over all cells $j$ signaling to cell $i$, and denote this average as $v_i$. We then set the input to the $i$th cell to $u_i = h(v_i)$, where $h(\cdot)$ is the signaling relation programmed into the computer. To enact mutual inhibition, increasing gene expression in one cell must decrease gene expression in neighboring cells. Therefore, since higher-intensity light induces higher gene expression, we select $h(\cdot)$ to be decreasing.

We chose a grid layout with periodic boundary conditions in which each cell signals four other cells reciprocally. This layout satisfies all assumptions discussed in the subsection entitled Summary of theory, therefore we can predict contrasting patterning in a full system of $N$ cells based on the bistability of an equivalent 2-cell system. If the 2-cell system is monostable, then both cells express the same level of gene, and the $N$-cell system also has a stable state in which all cells express the same level of gene. Inversely, if the 2-cell system is bistable, then one of the stable states corresponds to one cell expressing high levels of gene and the other, low, and the other stable state corresponds to the opposite situation. In this case, two stable, contrasting steady-state patterns also exist for the $N$-cell system; that is, one subset of the $N$ cells expresses identically high levels of gene, and the remaining cells express identically low levels of gene (or vice versa). Contrasting steady states can be visualized as checkerboards in which neighbors alternate between high and low. The 2-cell system can be assessed for bistability using a standard technique illustrated in Fig. 2 and described in Supplementary Section 1.

To apply the theory, we must know (1) the dose response, in our case in vivo gene expression levels under varying intensities of light; and (2) the form of the signaling relation, here programmed in silico. Thus, to carry out lateral inhibition experiments, we needed to measure an empirical dose response of cells to light, and define the computational signaling relation controlling light inputs such that intensity was inversely related to the responsiveness of cells interacting with the target.

## Summary of theory.

Consider a system of $N$ identical cells modeled as single-input, single-output dynamical systems. Biochemical concentrations $\mathbf{x}_i(t) \in \mathbb{R}_+^n$ in the $i$th cell evolve

according to

$$\frac{d}{dt}\mathbf{x}_i(t) = f(\mathbf{x}_i(t), \mathbf{u}_i(t)).$$

Each cell has output $w_i(t) = g(\mathbf{x}_i(t)) \in \mathbb{R}_+$ and input $u_i(t) \in \mathbb{R}_+$. Let the vector $\mathbf{x}(t) \in \mathbb{R}_+^{Nn}$ be the vertical concatenation of the vectors $\mathbf{x}_i(t)$ for all $N$ cells, and similarly for $\mathbf{w}(t) \in \mathbb{R}_+^N$ and $\mathbf{u}(t) \in \mathbb{R}_+^N$. We assume each cell has a static input–output characteristic $T(\cdot)$, that is, if a cell is given constant-in-time input $u_i(t) = u_i^\dagger$, it will reach a globally asymptotically stable hyperbolic equilibrium $x_i^\dagger$ solving $0 = f(\mathbf{x}_i^\dagger, u_i^\dagger)$ with output $w_i^\dagger = T(u_i^\dagger) = g(\mathbf{x}_i^\dagger)$. We assume $T(\cdot)$ is bounded and increasing, meaning that increasing the input increases the output. In our setup, the static input–output characteristic corresponds to the empirically measured dose response.

Suppose the outputs of cells are connected to the inputs of other cells, forming a network. We capture information about which cells signal to which by way of the interconnectivity matrix $M \in \mathbb{R}_+^{N \times N}$ with entries $[M]_{ij} = 0$ if cell $j$ does not signal to cell $i$ and $[M]_{ij} > 0$ otherwise, with the value $[M]_{ij}$ indicating the strength of signaling. We require that the sum over all entries in a row equal the same constant, $\mu \in \mathbb{R}_+$, regardless of the row, i.e., $\sum_j [M]_{ij} = \mu$ for all $i$. In our setup each cell receives signals from four other cells with equal weights $\frac{1}{4}$, therefore $\mu = 1$. Defining $\mathbf{v}(t) = M\mathbf{w}(t)$, we model lateral inhibition by letting the input to cell $i$ be given by $u_i(t) = h(v_i(t))$, where $h(\cdot) : \mathbb{R}_+ \to \mathbb{R}_+$ is bounded and decreasing.

**Model reduction theorem.** Let $\mathbb{1}_m$ represent the length-$m$ column vector of all ones, and similarly for $0_m$. If there exists a matrix $\overline{M} \in \mathbb{R}_+^{2 \times 2}$ such that

$$ML = L\overline{M} \text{ where } L = \begin{bmatrix} \mathbb{1}_m & 0_m \\ 0_{N-m} & \mathbb{1}_{N-m} \end{bmatrix} \quad (1)$$

for some indexing of cells, then $\overline{M}$ is an interconnectivity matrix for an equivalent 2-cell system whose steady-state solutions correspond to steady states of the $N$-cell system with interconnectivity matrix $M$. In other words, if $\overline{\mathbf{w}}^* \in \mathbb{R}_+^2$ is the output corresponding to a steady-state solution to the 2-cell system, then $\mathbf{w}^* = L\overline{\mathbf{w}}^*$ is a steady-state output to the $N$-cell system. $\square$

Note that the cells indexed 1 through $m$ take on steady-state output values $\overline{w}_1^*$ while those indexed $m + 1$ through $N$ take on steady-state output values $\overline{w}_2^*$. Condition (1) is satisfied when cells can be grouped into two subsets within which nodes are interchangeable; that is, reindexing nodes within a subset will not change $M$[41].

When

$$\overline{M} = \begin{bmatrix} 0 & 1 \\ 1 & 0 \end{bmatrix},$$

the steady states of the 2-cell system are determined graphically from the fixed points of $h(T(h(T(\cdot))))$, as shown in Fig. 2 and explained in Supplementary Section 1. For the reduced 2-cell system the graphical test also ensures stability of the points corresponding to the lower/upper intersections in Fig. 2, and instability of the point corresponding to the middle intersection, when the cellular dynamics are monotone in the input/output sense[49]. The stability properties established graphically for the 2-cell system are preserved in the full $N$-cell system when additional assumptions hold. Our setup satisfies one such assumption from ref. [40], which stipulates cells within a subset not signal to each other. Thus, if $\overline{\mathbf{w}}^*$ is the output corresponding to a stable state in a bistable 2-cell system, then in the $N$-cell

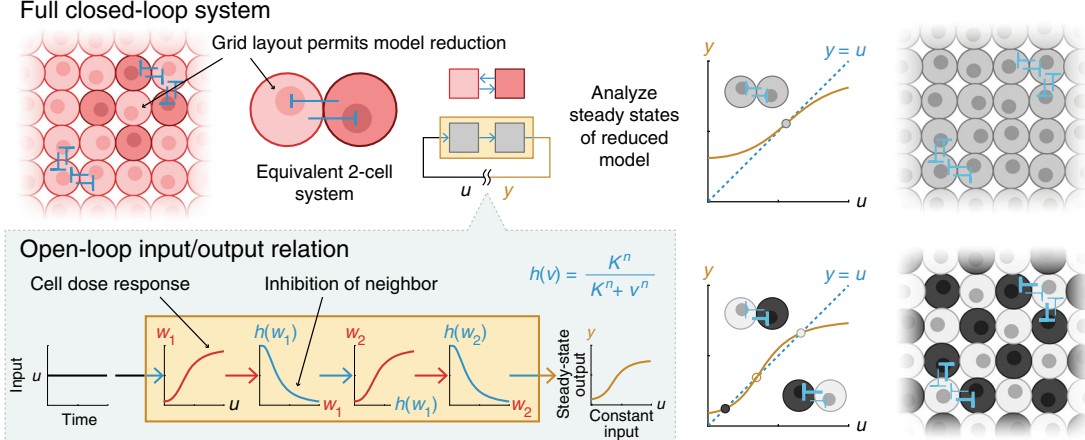

**Fig. 2 Spontaneous contrasting patterning in a large lateral inhibition system can be predicted from bistability in a 2-cell system.** The bistability of a 2-cell system can be used to evaluate contrasting patterning in a full network of isogenic, mutually inhibiting cells arranged in a grid with periodic boundary conditions. Bistability of the corresponding 2-cell system implies contrasting patterns exist in the full system (see subsection entitled Summary of theory). Bistability can be assessed through an analytical test in which the closed-loop dynamical system is opened into an input/output system by breaking the feedback loop. If a constant-in-time input to the open-loop system produces a steady-state output value equal to the input value, then that value is a steady state for the corresponding closed-loop system.

system, cells in one subset have higher output than cells in the other subset. If the cells belonging to different subsets are spatially interlaced or alternating, then the high/low dichotomy produces a spatially contrasting pattern such as a checkerboard.

**Empirical characterization informs computational parameters.** We combined the blue light-inducible VP-EL222 expression system[38,50] with a fast-acting nuclear translocation reporter (dPSTR)[39] to control and measure gene expression in *Saccharomyces cerevisiae*. In the dark, constitutively expressed red fluorescent protein (RFP) fused to the synthetic bZip domain SZ2[51] is equally distributed between nucleus and cytoplasm due to passive diffusion through the nuclear membrane. Under exposure to blue light, VP-EL222 molecules dimerize and bind the cognate promoter to activate expression of a protein comprising two nuclear localization signals (NLS) and SZ1[51]. This protein then forms a heterodimer with the RFP reporter, thereby localizing fluorescence in the nucleus. We quantitated the degree of nuclear localization (nuclear localization score) as the difference between mean cytoplasmic and mean nuclear fluorescence normalized to the mean fluorescence across the entire cell. In principle, the score is 0 if cells are not at all responding (there is no nuclear localization) and positive otherwise (Fig. 3a, b).

We characterized the dose response of individual cells to constant, targeted blue light exposure (Fig. 3b, c). On average, cells exhibited a graded response to light intensity well described by a Hill function (Fig. 3c). Variability from cell to cell was greater than for individual cells across time, perhaps owing to variation in cell cycle state[52]. As the theory is deterministic, for patterning experiments we ultimately substituted single cells with computationally defined patches of 4 or 6 cells, with the patch response determined as the average response of the constituent cells. Generating score distributions for such patches by bootstrapping from the single-cell dose response data shows reduced temporal and patch-patch variability as well as reduced difference between temporal and patch–patch variability relative to the single-cell case (Fig. 3d, Supplementary Fig. 3).

Based on the range of cellular response scores, we defined the signaling relation $h(\cdot)$ for use in patterning experiments, which determined the light input administered to a patch as a function of the average scores of neighboring patches at each time step. We

chose an inhibiting Hill function with fixed Hill coefficient $n = 2$ and a single free parameter $K$ with smaller values corresponding to sharper inhibition. We combined the empirical dose response with the computational signaling relation to generate theoretical predictions for the mono- or bistability of a 2-cell lateral inhibition system as $K$ was varied between 0 and 1, corresponding to non-patterning or patterning outcomes in a full system.

**Cell-in-the-loop generates spontaneous checkerboard patterns.** We ran a series of patterning experiments emulating lateral inhibition. Cells were randomly assigned to patches such that cells belonging to the same patches were not necessarily neighbors in physical space, thereby reducing spurious correlations that might arise from spatially dependent factors other than the targeted light input. Once assigned, cells remained in the same patch throughout the duration of an experiment. Patches were arranged to neighbor each other in virtual space as visualized on a checkerboard (Fig. 4).

During patterning experiments, we took measurements and calculated new light inputs every 10 min, about the same rate as the estimated time constant for cell response to a switch in illumination intensity (Supplementary Section 2.2). This choice allowed us to avoid adjusting inputs more frequently than responses to the previous input could be detected. We tested systems of 16 patches with 6 cells per patch for four values of $K$ between 0.1 and 1. Spontaneous patterning was always achieved in the $K = 0.1$ case and never in the $K = 1$ case, with mixed results for $K = 0.2, 0.3$, near one of the theoretically predicted critical points (Supplementary Fig. 5a). Sample time traces at $K = 0.1$ and $K = 1$ show, respectively, the gradual deviation in score between sets of alternating patches that characterizes a contrasting pattern, or a rapid adoption of a non-patterning state. Visualizing the checkerboard at individual time points or averaged over the last hour clearly depicts the distinction between the two cases (Fig. 5a). In the bistable case, which of the two possible checkerboards emerged in a given experiment depended on the stochastic initial conditions and continued noisy influences during system evolution (Supplementary Fig. 5).

When averaged over the last hour and across experiments, the contrast level (mean scores of sets of alternating patches) was quantitatively well predicted by theory in the bistable region and

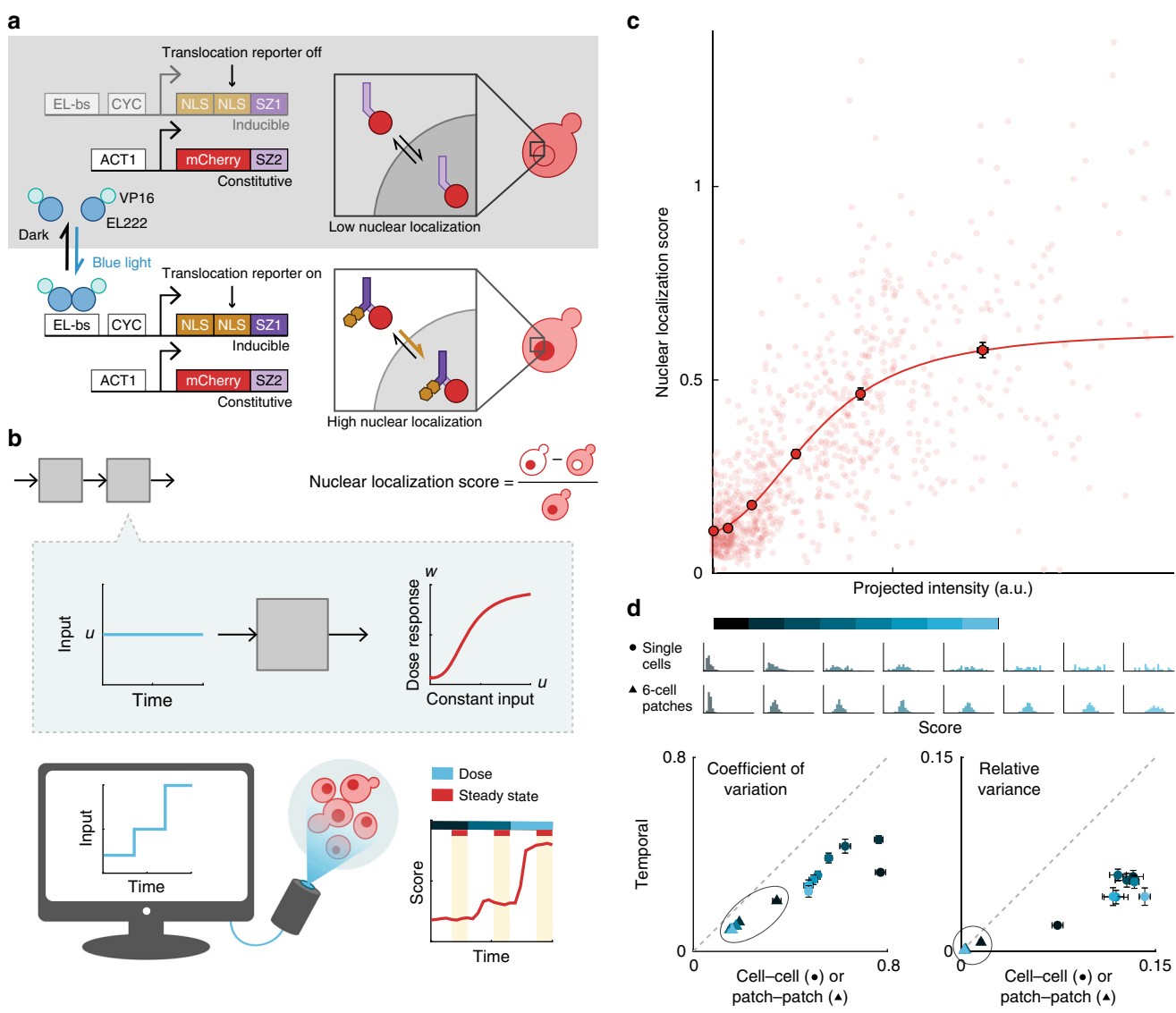

**Fig. 3 Experimental characterization of cellular dose response curves shows population-level gradedness. a** Blue light induces gene expression in optogenetically responsive *Saccharomyces cerevisiae* via the VP-EL222 system[36], as measured by a nuclear translocation reporter (dPSTR)[39]. Cells constitutively express RFP fused to SZ2. In the dark, RFP is distributed equally between nucleus and cytoplasm due to diffusion. Blue light induces VP-EL222 dimerization, which activates expression of an NLS fused to SZ1. SZ1 binds SZ2, such that the NLS promotes localization of RFP in the nucleus, increasing nuclear fluorescence relative to cytoplasmic fluorescence. **b** For a single dose, individual cells were illuminated for 80 min with constant-intensity light. Single-cell responses to a single intensity were calculated as the average score over the last 40 min. Pictured in the schematic are average input intensities and responses for ~700 cells for three consecutive doses. **c** Although individual cell responses vary, on the population level nuclear localization is graded with respect to the intensity of the light input. Responses of single cells to single doses were pooled across three experiments (no outline). A Hill function was fit to the quantile means (180 or 181 samples per quantile) to generate the final dose response (solid outline). Source data are provided as a Source Data file. **d** Grouping cells into patches reduces both cell–cell and temporal variability. Scores were binned by projected intensity (blue shaded bar) to generate histograms of score distributions for given input levels (samples per bin left to right: 344, 183, 69, 108, 102, 57, 35, and 26). Bin cutoff is twice the maximum projected intensity used in final patterning experiments. Histograms for 6-cell patches were generated by bootstrapping from individual cell scores within a bin for a total of 200 bootstrapped patches per intensity level. For an ergodic process, cell–cell and temporal variation would be equal (gray dotted line); here, responses appear to be more variable from cell to cell than for single cells across time. Grouping cells into patches of six across which response scores are averaged (triangles) reduces the magnitude of difference between cell–cell or patch–patch and temporal variability. Error bars are s.e.m.

the overall brightness (mean score across all patches) was well predicted in the monostable region (Fig. 5b). When considering individual experiments, the variability in overall brightness increased with increasing $K$. In the predicted monostable cases, stochasticity also introduced a difference between the means over alternating patches, though statistical analyses confirm that the difference was indistinguishable from random (Supplementary Table 4). Taken together, these results suggest that the

deterministic theory calibrated to population averages is an excellent quantitative predictor for mean system behavior across time, patches (cells), and experiments, while at the same time even small amounts of cell–cell variability and temporal stochasticity may cause a given experiment to deviate considerably from quantitative forecasts.

Because our setup allowed us to monitor both gene expression levels and cell signaling levels, we were able to assess convergence

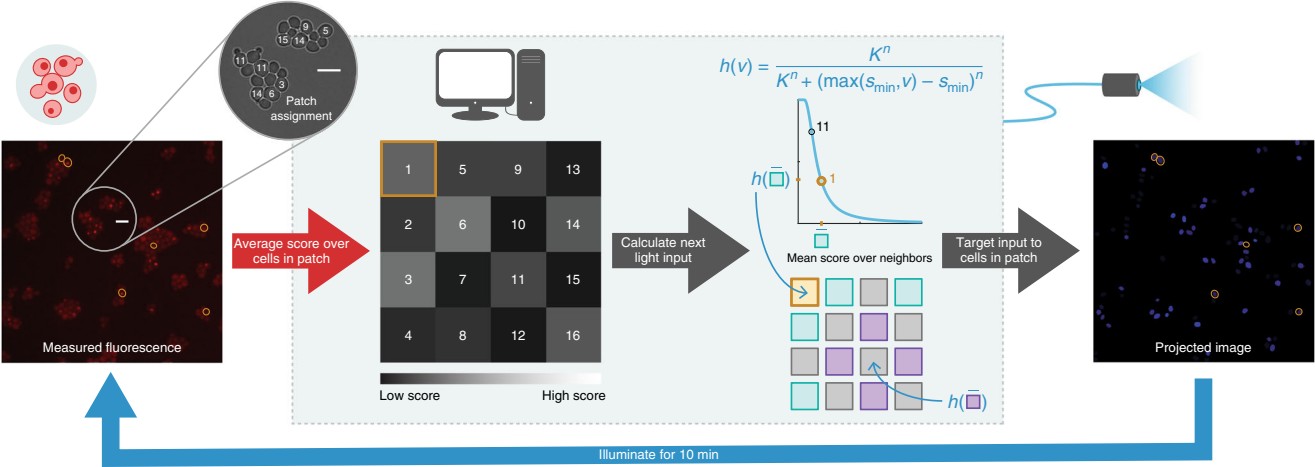

**Fig. 4 Schematic of automated workflow for patterning experiments.** Cells growing in a monolayer were placed under the microscope, which was connected to a computer system that controlled the camera and blue light projection system. Cells were segmented and tracked in brightfield images and scored in fluorescence images (here, false-colored red). After the first image was acquired, cells were randomly assigned to patches to which they belonged for the remainder of the experiment. The grid visualizes scores at a single time step, with each square representing one patch of cells. The brightness of a square corresponds to the average score of the constituent cells in the corresponding patch. The signaling relation, defined before experiment start, determined the input light intensity administered to patches for the next time step (10 min) based on the scores of neighboring patches to the north, south, east, and west, with periodic boundary conditions. The relation $h(\cdot)$ was chosen to be an inhibiting Hill function with a minor correction to better utilize the range of available illumination intensities (i.e., a patch received maximum illumination intensity if the average score over its neighbors was at or below $s_{min} = 0.05$). The Hill coefficient was fixed at $n = 2$ for all experiments. The bifurcation parameter $K$ was fixed for each experiment at a value between 0.1 and 1. Cells in the same patch received the same input intensity targeted individually to each cell, as shown in the projected image. Scale bar in fluorescence and brightfield images is 10 μm.

to (quasi-)steady state by comparing instantaneous input-output curves (patch score vs. administered intensity) to the steady-state dose response (Fig. 5c). Specifically, since the time for cells to converge to steady state under exposure to light of constant intensity (40 min) was longer than the time between changes to input intensity during experiments (10 min), the instantaneous input–output curve during a patterning experiment would only match the empirical dose response curve if the administered intensity remained relatively constant for several frames before a given time point—i.e., if there was little temporal variability for at least 40 min preceding the frame. Directly plotting the temporal variability in administered input to individual patches does indeed reveal a decrease from the first to the last experimental hour regardless of $K$ value (Supplementary Fig. 6a).

Lastly, we examined the effect of patch number on patterning outcomes through four experiments with 36 patches, 4 cells per patch, and $K = 0.1$. None of the experiments spontaneously achieved a checkerboard pattern across the whole board in 3 h, although a control experiment preinduced with the pattern showed that it was indeed persistent (Supplementary Fig. 7). The input/output curve did not approach the empirical dose response (Supplementary Fig. 8) and temporal variability in administered intensity was the same during the first and third experimental hour (Supplementary Fig. 6b), further supporting the conclusion that the system never reached steady state. Interestingly, one experiment produced two checkerboards in opposite corners that persisted throughout the last experimental hour, but were inverted relative to each other and did not resolve before the end of the experiment (Supplementary Fig. 5b). Other experiments also exhibited transient local patterning, although to a lesser degree. The local patterning and the increased convergence time are consequences of the fact that a 36-patch system admits a much larger space of possible configurations than a 16-patch system. Although variability in 4-cell patches was only modestly larger than in 6-cell patches (Supplementary Fig. 3), the stochasticity may also have contributed to a longer convergence

time. These and related challenges will require further investigation in future efforts to synthetically generate gene expression patterns with single-cell granularity.

## Discussion

In this work, we employed cell-in-the-loop, a closed-loop, hybrid in vivo/in silico approach, to validate a theory for spontaneous gene expression patterning among laterally inhibiting cells. We engineered *S. cerevisiae* to respond optogenetically to light inputs, then emulated cell-to-cell signaling in real time by modulating the intensity of light inputs to cells based on real-time measurements of gene expression. The theory made accurate quantitative predictions for average steady-state patterning outcomes across a range of parameters. Increasing system size—by increasing the space of possible dynamic behaviors—diminished the probability of achieving global patterning on short timescales in the absence of initial or external bias. Further thereotical research should explicitly incorporate cell–cell variability and temporal stochasticity in order to improve our understanding of variation in individual experimental outcomes, patterning robustness, and the link between individual-level and population-level behavior.

Prior work using optogenetics to generate persistent spatial patterns in living cells has focused on reproducing[53–55] or processing[56] pre-existing images projected by the light input. In comparison to these studies, light in our system does not a priori encode a pattern to which the cells conform; rather, light acts as a virtual signal transmitted from cell to cell. The input intensity is determined by cellular responses that are in turn influenced by the received intensity, establishing a closed-loop relationship independent of external control. That similar patterns are ultimately observed in both the cellular responses and the optogenetic inputs arises as a consequence of their mutual dependence.

Depending on the application, cell-in-the-loop offers benefits over purely biochemical or purely computational approaches. First, it reduces the number of components that must be engineered into cells. We integrated a single optogenetically induced promoter and

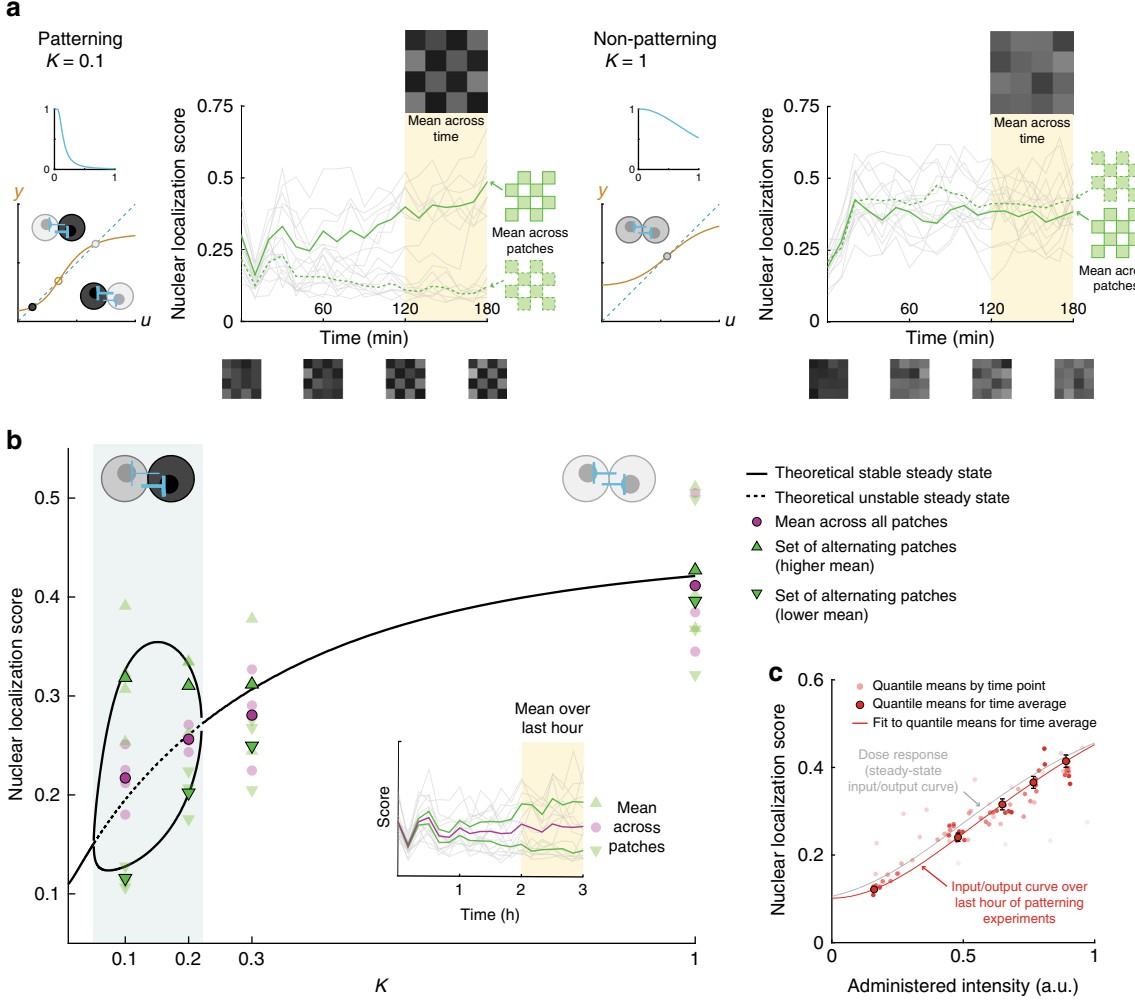

**Fig. 5 Theory quantitatively predicts spontaneous patterning and patch intensity during cell-in-the-loop experiments. a** Sample time traces show emergence of a contrasting pattern ($K = 0.1$) or convergence to a non-patterning state ($K = 1$). Gray lines correspond to score traces for individual patches; green lines indicate the mean scores of sets of alternating patches. Checkerboards visualize scores at single time points (bottom) or averaged over the last hour (top). **b** Theory quantitatively recapitulates experimental results for mean patch response score. Black lines denote theoretical steady-state points as a function of the bifurcation parameter $K$, with solid indicating stable and dashed, unstable. All points are averaged over the last hour. Faded points correspond to individual experiments; solid outlines are averages across experiments ($N = 4$ for $K = 0.1$, $N = 3$ for $K = 0.2$, 0.3, 1). Magenta circular points are averages over all patches. Green points are averages over sets of alternating patches, with upward- and downward-facing triangles denoting the higher and lower of the two means respectively. Stochasticity in the experimental system introduced contrast between the average means of sets even in regions that were deterministically suggested to be monostable. As predicted, the contrast level (difference between means of sets of alternating patches) was higher for lower $K$. Experiment-to-experiment variation in overall brightness (score averaged across all patches) was greater for higher $K$, an effect that cannot be accounted for in a purely deterministic theory. **c** The 16-patch system with six cells per patch converges to a steady state by ~2 h into patterning experiments. Solid outline, score values averaged over the last hour for individual patches and split into quantiles by administered intensities show decent agreement with the empirical steady-state dose response curve. Error bars are s.e.m. Circles without outlines are quantiles at individual time points pooled from $N = 13$ experiments (darker red at later times). For comparison, the dose response curve fit to empirical data is plotted in gray. Source data are provided as a Source Data file.

a single reporter, and were able to modulate patterning outcomes simply by reprogramming the computer. In this way, we circumvented issues associated with synthetic cell-to-cell signaling, including parameter matching and crosstalk[28,57], and alleviated complications such as burden[58,59] that arise from integrating complex networks into cells[29,31]. We were also able to achieve spatiotemporal control over the whole population of interacting individuals and probe stochasticity at a finer level than would be attainable with a conventional biochemical implementation.

Compared to a computer simulation, cell-in-the-loop makes no assumptions about cell behavior or the form of biological noise, since the cells themselves are incorporated into the system. Although we used this setup to test the validity of a theoretical

principle, one could also envision testing the accuracy of a full model for cell-to-cell signaling by simulating a proposed physical mechanism of interaction, then comparing the outcome of such a system to the outcome of a purely physical system. Cell-in-the-loop also allows one to track system components that might otherwise be inaccessible or difficult to measure. For example, we were able to monitor the levels of both gene expression and virtual signal simultaneously, which could be difficult to achieve in a solely biochemical setup.

Once established, a cell-in-the-loop system could couple with more complex cellular processes to achieve real-world results. One could also envision using cell-in-the-loop as a rapid prototyping platform or stepping stone to a fully biochemical implementation.

In this paradigm, one would begin with minimally engineered cells and then sequentially replace in silico components with biochemical ones, testing at each stage whether the remaining portions of the network ought to be modified in structure or value before the next component is incorporated into the cell. As an illustration, suppose one were to implement lateral inhibition with the basic transcriptional repressor circuit from[27], in which cleaved intracellular Notch domain activates a repressor for expression of Delta, the target molecule of Notch recognition domain. One might employ the synNotch toolbox for mammalian cells, which includes as design choices the recognition domain/target molecule pair and the transcription factor constituting the intracellular synNotch domain[25]. One might begin with a cell-in-the-loop approach by engineering just the repressor under optogenetic control, and define computational variables for the concentrations of recognition domain, transcriptional activator (cleaved synNotch intracellular domain), and target molecule (plus intervening pathway components) in each cell. Using calculated activator concentration to modulate the light input, one could simulate different parameters for the remaining components to decide upon an activator. One could then engineer a circuit with repressor controlled by the chosen activator and place the activator under optogenetic control. With another round of cell-in-the-loop, one could further restrict the range of parameters for the combination of target molecule and recognition domain that would produce the desired effect in vivo, and use these numbers to guide construction of the final biochemical circuit in live cells.

Although the above example has relatively few components and therefore few design choices, the benefits of cell-in-the-loop should scale with circuit complexity as the parameter space—and therefore the potential number of full circuits to test—increases with each added component. The approach could also reveal shortcomings in proposed designs: for example, in our setup, 36-patch systems failed to produce spontaneous patterns on our experimental timescale even with perfect deterministic signaling, suggesting that it could be challenging to achieve large-scale lateral inhibition patterning biochemically in a similar context. Thus, the addition of cell-in-the-loop to the biological engineering process could greatly decrease the time, expense, and effort required to develop synthetic multicellular systems for an increasingly rich and promising array of applications.

## Methods

**Plasmid and yeast strain construction.** *Escherichia coli* TOP10 cells (Invitrogen) were used for plasmid cloning and propagation. The dPSTR reporter plasmid (pDB161) contains the coding sequences of UbiY-2xSV40NLS-SynZip1[39], expressed from an EL222-responsive promoter (p5xBS-CYC180)[38], and mCherry-SynZip2[39], expressed from the constitutive *ACT1* promoter. It was constructed by first replacing the promoter pRPL24A in the plasmid pDA183[39] by pACT1 using SacI-XbaI cut sites and subsequently replacing the promoter pSTL1 by P5xBS-CYC180 using PCR and SapI-based Golden Gate cloning[60].

The yeast strain used in this study (DBY165) was constructed by transforming the PacI digested plasmid pDB161 into DBY41[38], a strain with BY4741 background expressing VP-EL222[50] from the *ACT1* promoter. The transformation was performed using the standard lithium acetate method[61].

**Culture preparation.** Cells were grown at 30 °C in synthetic medium (SD) consisting of 2% glucose, low fluorescence yeast nitrogen base (Formedium), pH 5.8, 5 g/l ammonium sulfate, and complete supplement of amino acids and nucleotides. Cultures were started from plate, diluted, and maintained at $OD_{600} < 1.5$ between 24 and 32 h before an experiment. For each experiment, between 3 and 5 mL of cell culture were centrifuged at 20 °C, 3000 RCF for 6 min and enough supernatant was removed to achieve an approximate $OD_{600}$ of 4 after resuspension. Cells were then immediately placed on agarose pads, prepared according to the procedure in Supplementary Section 2.1.

**Imaging.** Images were taken under a Nikon Ti-Eclipse inverted microscope (Nikon Instruments) with a ×40 oil-immersion objective (MRH01401, Nikon AG, Egg, Switzerland), pE-100 bright-field light source (CoolLED, UK), and CMOS camera ORCA-Flash4.0 (Hamamatsu Photonic, Solothurn, Switzerland) water-cooled with a refrigerated bath circulator (A25 Refrigerated Circulator, Thermo Scientific). The

temperature was maintained at 30 °C by an opaque environmental box (Life Imaging Services, Switzerland), and a dark cloth was additionally placed over the microscope to fully shield cell samples from external light. Experiments were conducted with a diffusor and a green interference filter placed in the bright-field light path, with the Nikon Perfect Focus System (±5 AU) enabled. Fluorescence images were acquired using a Spectra X Light Engine fluorescence excitation light source (Lumencor, Beaverton, USA), filter cube with excitation filter 565/24 nm, emission filter 620/52 nm, and beam splitter HC BS 585 (AHF Analysetechnik AG, Tübingen, Germany). The final fluorescence images used for analysis were maximum projections across z-stacks of 5 images spanning 0.6 μm.

During experiments, the microscope was operated by the open-source software YouScope[62]. Cell segmentation and tracking were performed on brightfield images using software tools developed by Rullan et al.[36] based on Dimopoulos et al.[63] and Ricicova et al.[64]. For each cell, the mean fluorescence in the nucleus, cytoplasm, and across the entire cell were automatically calculated using custom Matlab® (MathWorks) scripts following the procedure in Supplementary Section 2.2.

**Light-delivery system.** Optogenetic inputs were delivered to cells using the setup developed in Rullan et al.[36], in which images generated on the computer are projected by a digital mirror device through a system of lenses that focuses the light onto a microscope slide. Two neutral density filters (Thorlabs, 25 mm absorptive, optical densities 0.5 and 1.3) were placed serially to achieve a total density of 1.8.

To ensure light mapped properly from the projector to the cell, images were modified prior to projection in order to map pixels on the DMD to pixels in the camera images. The mapping was determined through the procedure outlined in Rullan et al.[36], Fig. S6B. The procedure was performed immediately before experiment start on an area of the agarose pad unoccupied by cells.

We observed that there was a sigmoidal relationship between the administered illumination intensity and the measured illumination intensity in the projection images, and also that images of uniform intensity did not evenly illuminate the sample plane. To compensate for these effects, we modified images before projection to linearize the administered-to-measured intensity relationship and also to reduce the intensity of overilluminated regions to match the level attained by underilluminated regions. Details of the procedure are available in Supplementary Section 2.5.

Custom Matlab® code was used for manually calibrating projector intensity before experiment start (Supplementary Section 2.5) and for automatically carrying out experiments.

**Dose response.** The theory relies on a deterministic dose response curve in which the expected steady-state response score of a cell increases as a function of constant input intensity. We performed a series of dose response experiments to verify that these conditions held for our yeast strain and then calculated a dose response curve from the average response of cells to varying measured projected intensities.

Cells tended to respond much more strongly and unpredictably to the first administered input than to later inputs regardless of the intensity of the first input. Therefore, before administering any doses, all cells on the dish were illuminated for 10 min with uniform, middling intensity light, then left in the dark for 20 min to allow the response to decay. Multiple doses were then administered in immediate succession to cells on the plate. For a single dose, cells were illuminated with individually targeted light with constant administered intensity per cell. Cells were imaged every 10 min. Cells that were not successfully segmented and tracked at all sampled time points in a dose were discarded. The steady-state response of a single cell was calculated as the average of the scores from 40 to 80 min under illumination.

The final dose response curve was fit to data aggregated from three experiments. Individual cell responses were binned by projected intensity into 6 quantiles. The final dose response curve in the form of a leaky activating Hill function

$$f(x) = a + b\frac{x^n}{c^n + x^n}$$

was fit to the mean score values within each quantile vs. the mean measured projected intensity within that quantile.

**Patterning experiments.** Before experiment start, all cells on the plate were illuminated for 10 min with uniform, middling intensity light, then left in the dark for 10 min such that responses would not fully decay, allowing for some initial variation in nuclear localization score. Experiments lasted 3 h, during which cells were imaged and their inputs adjusted every 10 min. Preliminary experiments confirmed that cells were alive and responsive up to 6 h after placement under the microscope, although final experiments were constrained to 3 h to ensure cells remained in a monolayer.

*Patch construction.* Cells were randomly assigned to groups with a fixed number of cells per group. Each group corresponded to a single computationally defined patch. The score for a patch was calculated as the average of the scores of the cells comprising the patch. By administering an input to a patch, we mean the cells in that patch were individually targeted with the same administered input. Most experiments were performed with 16 patches of 6 cells per patch. Higher-dimensional experiments were performed with 36 patches of 4 cells per patch.

Patches were arranged in virtual space into a square grid where each patch was connected to (interacted with) each of the patches to the north, south, east, and west, with periodic (wrap-around) boundary conditions such that each patch interacted with four other patches (its neighbors). Note that, because cells were randomly assigned to patches, cells to neighboring patches in virtual space were not necessarily adjacent in real space.

*Signaling relation.* Every imaging period (10 min), the input to each patch was adjusted according to the signaling relation, which was chosen to be a Hill function with a minor computational adjustment to better utilize the available range of illumination intensities. Specifically, the relation was given by

$$h(v) = \frac{K^n}{K^n + (\max(s_{\min}, v) - s_{\min})^n} \quad (2)$$

where $v$ was the average score across all four neighbors of a patch, $n$ was fixed at 2, and the minimum score $s_{\min} = 0.05$ was the cutoff to deliver maximum illumination intensity. The bifurcation parameter $K$ was fixed within a given experiment at a value between 0.1 and 1, with higher $K$ corresponding to weaker repression.

In particular, let $x_k^i$ be the score of the $i$th patch in the $k$th frame, and $u_k^i$ be the input to the patch between the $k$th and $(k + 1)$th frames. Then $u_k^i$ was calculated as

$$u_k^i = h\left(\sum_j x_k^j\right)$$

where the summation is taken over the neighbors of patch $i$ and $L(\cdot)$ is as given in Eq. (2).

**Reporting summary**. Further information on research design is available in the Nature Research Reporting Summary linked to this article.

## Data availability
The cell score data underlying Figs. 3c and 5, as well as Supplementary Figs. 5, 6, and 8, are available in the Source Data file. Remaining data, plasmids, and strains are available upon request.

## Code availability
Custom code to run experiments was closely integrated with the experimental setup and cannot be executed without associated hardware. Data analysis reported in this work was performed on the raw data using standard Matlab® functions (see the data and sample script in the Source Data file). Code is available upon request.

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

## Acknowledgements

We would like to thank Marc Rul·lan for his help with the setup and guidance throughout the project, as well as Brian M. Lang for his advice on statistics and general linear modeling for the dose response data. This project received funding from the United States Air Force Office of Scientific Research (AFOSR) grants FA9550-14-1-0089 and FA9550-18-1-0253, the European Research Council (ERC) under the European Union's Horizon 2020 research and innovation programme (CyberGenetics; grant agreement 743269), FET-Open research and innovation actions grant under the European Union's Horizon 2020 research and innovation programme (CyGenTiG; grant agreement 801041), and Swiss National Science Foundation (SNSF) grant 316030_177079. Publication made possible in part by support from the Berkeley Research Impact Initiative (BRII) sponsored by the UC Berkeley Library.

## Author contributions

M.K., M.L.P., and M.A. conceived the project and designed the study with initial input from D.B. M.L.P. and D.B. planned the experiments. D.B. engineered yeast strains. M.L.P. performed the experiments and analyzed data. M.L.P. wrote the manuscript with contributions from all authors, and all authors contributed to editing the final manuscript. M.A. and M.K. supervised the study and secured funding.

## Competing interests

The authors declare no competing interests.
