## [Peer Review File · Nature Communications]

Reviewers' Comments:

Reviewer #1:

Remarks to the Author:

The manuscript of Perkins and collaborators deals with the development of a theoretical-experimental framework and approach for the control and study of emerging spatial patterns in living cells. The so-called cell-in-the-loop approach experimentally validates a theoretical framework for the establishment of checkerboard-type cell patterning triggered by cell to cell lateral inhibition. The bistability of the system is characterized as a closed-loop dynamic system. The model predicts quantitatively the patterning outcomes across a space of parameter values. To experimentally validate these predictions, the closed-loop is broken/opened into an open-loop system (input/output). Bilayers of yeast cells are equipped with an optogenetic switch controlling gene expression of an easy to track visual output (FP). Cell-cell signalling is emulated by the spatially resolved application of the input, namely illumination of individual groups of cells at a given intensity, for designated periods of times. Finally, the emergence of the expected pattern is evaluated and analysed in relation to the range of parameters considered. Overall this approach represents a step forward for the design and implementation of synthetic signalling networks in living cells. Synthetic biology has already proven its potential towards constructing systems of high complexity by rationally combining engineered biological modules based on a rational design. However, most of current strategies for the engineering of multicellular patterning, be it microbial communities or higher-level structures as in animal organs/tissues, lack actually the quantitative, theoretical support needed for performing powerful predictions to support such rational design. This work precisely contributes closing the gap between exclusively theoretical studies and the actual engineering of the biological system, what is mostly welcome. It represents one of the first steps towards truly being able to rationally design and constructs molecular and cellular biological systems. It is a very novel approach in its nature and will find fast application in several fields of research, ranging from the understanding of the principles ruling complex cellular pattern formation in every domain of life up to the development of biotechnological applications such as production of synthetic organs, biomaterials and bacterial biofilms/communities for metabolite production. The manuscript is very well written and clear. The figures are of high quality and the content/results/discussion adequately represented. I therefore support its publication with revisions. Some issues need to be reworked-clarified.

It is not clear to me from the data shown and description of the system what is the parameter space of the starting system that would lead to an initial breakage in the stability which will then be reinforced by the external input thereon. In other words, what is the threshold of signal difference of the initial stochastic distribution of cells/group of cells (Δ among cells) that needs to be overcome in order for the model to instruct the activation of the input signal. Is there any situation in the modelled system in which the system remains monostable? What about in nature? Some examples, comparisons, discussions thereof are needed to illustrate the reader.

In relation to the previous point, in figure 5 a, two situations corresponding to extreme K values are shown. The initial states are different, and considering that these initial states arise stochastically (no effect of the K prior to this time point), how does the model operate on them differentially? Could it be that two different initial states give rise to different outcomes despite the same K value is operating? Conversely, given the same initial state, how do the "Ks" operate, and what is the parameter space that leads to the bifurcation?

Does the model consider cell growth and/or division? Would this be difficult to implement into it?

What were the experimental considerations in terms of rates of transcription/translation, thresholds thereof for application of the different illumination regimes (durations/time steps). It should be stated how and why the actual set ups were selected.

To extrapolate to more complex networks in vivo, the next logic step would be to couple the

output to a signal that has a negative effect on the neighbouring cell, for instance that leads in a proof of principle to reduction of the output signal of a cell already taken as in the other state. In principle this would lead to a faster increase in the delta, and thus emergence of the pattern. What modules should incorporate the model to include this. What are the implications in terms of kinetics of the system? This is representative to what it happens in several systems occurring in nature.

How robust are the systems? Is there any proxy of it? What happens when as in natural systems cells die, signals disappear, etc. Does the model incorporate this into account? This is particularly interesting in virtue of the limitations observed when the complexity of the system increases (e.g. 36 patches).

Overall, to increase the relevance for the general readership it would be interesting and useful to have clear examples of natural systems corresponding to the systems described in the article. In the same line, provide ideas, descriptions and discussion of putative engineered switches and cell synthetic signalling networks that could be developed in order to reconstruct the closed-loop system leading to bistability achieved by lateral inhibition in nature. A deeper and more complete discussion and perspectives should be included.

Reviewer #2:

Remarks to the Author:

I commend the authors on a very interesting paper that tackles a crucial problem in the emerging field of pattern formation. Namely, how a "cell-in-the-loop" approach can be used to control pattern formation. Detailed comments are attached.

Comments on:

“Cell-in-the-loop pattern formation with optogenetically emulated cell-to-cell signaling”

The manuscript reports the results of using a “cell -in-the-loop” approach for spatiotemporal control. Guided by theory, the authors engineered *S. cerevisiae* with a fluorescence reporter and emulated cell-cell signaling in *in-silico* via optogenetic control and showed that a virtual checkerboard pattern could emerge as a bifurcation parameter was varied. The results were in quantitative and qualitative agreement with the theory. This work stands as one of the first successful attempts to apply mathematical theory experimentally for multicellular patterning. The problem formulation is clearly stated and sufficient background and motivation is provided. The procedures and methods are clearly explained and sufficient details are provided for repeatability. The supplementary material complements the main text well and addressed a number of my questions.

Novelty and value to the field: This approach elegantly addresses many of the issues currently associated with spatiotemporal control. Primarily it bypasses having to construct explicit models of the intracellular and cell-cell dynamics that pure simulations require. This is advantageous, considering that the full structure of these models is usually unknown along with many of the parameters. Furthermore, simulations to these models may carry huge computational costs when individual cells are accounted for in a large population. By restricting cell-cell communication solely through user specified signals that are generated *in-silico*, this methods avoids issues that arise from crosstalk, stochastic effects, and burden on the cell from supporting large synthetic networks. Alleviating these unwanted interactions provides more reliable and replicable results. Through their implementation the authors had full access to the cell inputs and outputs, which is not something that is typically available in a pure biological experiment. This information provides insight that can be used in design or system identification. Even though only discussed briefly, this work provides insight on the role of stochastic effects in multicellular patterning and the limits where a deterministic model provides sufficient predictive power.

Overall, the manuscript is recommended for publication after the comments below are addressed. Major Comments:

- Is there a particular reason the closed loop input to individual cells is in the form $h(\sum_i w_i)$ rather than $\sum_i h(w_i)$? The latter seems more biologically relevant.
- Figure 3.c: Why are the error bars so small considering the large spread in the data?

- Figure 3.c: How is it possible to have a localization score greater than one?
- Figure 3.d and Figure S1.3: The variability between the 6-patch cells and the 4-patch cells seem quite similar. This is discussed briefly at the end of section 3.3 and it is claimed that patterning is not achieved due to the increased space of probabilities and not so much stochastic effects. Would it be possible to try the 36 patch test with 6 cells per patch just to rule this out?
- Results from Figure 3: Is it possible to estimate the time constant it takes for the cells to reach steady state? If so, how close is this to the 10 minutes at which the input was adjusted in the closed loop case? If larger, what are the pros and cons of increasing the input adjustment time?
- The theory assumed all cells to have the same I/O properties to render the analysis of the 2-cell system sufficient for the full system (correct?). In experiments, this would correspond to each patch having identical I/O properties. If the dose response is repeated at the patch level, will the I/O responses be similar enough (6 cells per path, 4 cells per path)?
- Is it true that if $n = 1$ in $h(\cdot)$, then one does not expect patterning regardless of K ? If so, would it be worth showing that the experiment is consistent with the theory?

Minor Comments:

- Page 8: In the first paragraph of section 3.3, a reference to Figure 4a is made even though it doesn't exist.
- Figure 3.d: label the x-axis (cell-cell variation?)
- Figure 4: Perhaps is best to define the variables in the $h(v)$ equation
- In 5.4 it is said that cells respond unpredictably to the first light dose and thus before any experiment they are illuminated for 10 minutes and then left in the dark for 20 minutes. Do we know why they respond this way? Also, how critical is the time span of the initial illumination (the 10 minutes)? Can varying this change any of the results?
- It is not clear how the periodic boundary conditions were enforced. Specifically during early time.
- Did you test whether indeed you could illuminate a single cell without exciting any of its neighbors?

Response to reviewers for “Cell-in-the-loop pattern formation with optogenetically emulated cell-to-cell signaling”

December 19, 2019

We are grateful to the editor and two anonymous reviewers for their constructive feedback. We believe our manuscript has greatly benefitted from the suggestions they provided.

The revised manuscript contains the following major changes:

1. Empirically calculated the time constant for cell responses (Sections 3.3 [Results] and S1.2.2). Provided analysis of patch-level dose responses (Section S1.3).
2. Incorporated a more detailed discussion of how the input adjustment time was chosen (Section S1.4.1).
3. Proposed a scheme for designing a synthetic lateral inhibition circuit using cell-in-the-loop as a stepping stone (Section 4 [Discussion]).
4. Incorporated examples of lateral inhibition in natural systems with associated citations [42-48] (Section 2 [Introduction]).
5. Fixed the axis limits for the dose response on Figure 3c (the previous version had unintentionally obscured one of the quantile points).

We have included a version of the manuscript with all changes from the previous draft highlighted in red text (“article_changes_marked.pdf”).

Below we have copied the comments from reviewers in black and our responses in blue underneath.

1 Reviewer 1

The manuscript of Perkins and collaborators deals with the development of a theoretical-experimental framework and approach for the control and study of emerging spatial patterns in living cells. The so-called cell-in-the-loop approach experimentally validates a theoretical framework for the establishment of checkered board-type cell patterning triggered by cell to cell lateral inhibition. The bistability of the system is characterized as a closed-loop dynamic system. The model predicts quantitatively the patterning outcomes across a space of parameter values. To experimentally validate these predictions, the closed-loop is broken/opened into an open-loop system (input/output). Bilayers of yeast cells are equipped with an optogenetic switch controlling gene expression of an easy

to track visual output (FP). Cell-cell signalling is emulated by the spatially resolved application of the input, namely illumination of individual groups of cells at a given intensity, for designated periods of times. Finally, the emergence of the expected pattern is evaluated and analysed in relation to the range of parameters considered. Overall this approach represents a step forward for the design and implementation of synthetic signalling networks in living cells. Synthetic biology has already proven its potential towards constructing systems of high complexity by rationally combining engineered biological modules based on a rational design. However, most of current strategies for the engineering of multicellular patterning, be it microbial communities or higher-level structures as in animal organs/tissues, lack actually the quantitative, theoretical support needed for performing powerful predictions to support such rational design. This work precisely contributes closing the gap between exclusively theoretical studies and the actual engineering of the biological system, what is mostly welcome. It represents one of the first steps towards truly being able to rationally design and constructs molecular and cellular biological systems. It is a very novel approach in its nature and will find fast application in several fields of research, ranging from the understanding of the principles ruling complex cellular pattern formation in every domain of life up to the development of biotechnological applications such as production of synthetic organs, biomaterials and bacterial biofilms/communities for metabolite production. The manuscript is very well written and clear. The figures are of high quality and the content/results/discussion adequately represented. I therefore support its publication with revisions. Some issues need to be reworked-clarified.

We thank the reviewer for their favorable assessment of our work and for their support for publication of our manuscript in *Nature Communications* with revisions.

1. It is not clear to me from the data shown and description of the system what is the parameter space of the starting system that would lead to an initial breakage in the stability which will then be reinforced by the external input thereon. In other words, what is the threshold of signal difference of the initial stochastic distribution of cells/group of cells (Δ among cells) that needs to be overcome in order for the model to instruct the activation of the input signal. Is there any situation in the modelled system in which the system remains monostable? What about in nature? Some examples, comparisons, discussions thereof are needed to illustrate the reader.

In our work we use “parameters” to refer to K and n in the computer-programmed signaling relation. The choice of bifurcation parameter K determines whether (a) the homogeneous state is unstable and the checkerboard states are stable (corresponds to bistability of the reduced system); or (b) the checkerboard states are unstable and the homogeneous state is stable (corresponds to monostability of the reduced system). In the former case, *any* amount of noise is sufficient to cause the system to pattern.

We varied K between experiments (not during experiments—clarified now in the caption to Fig. 4 and in Section 5.6.2). During experiments the light input was administered constantly to all cells, with intensity varying over time, rather than requiring a particular threshold for the activation of the input signal.

Please see the answer to the next question (2) for discussion regarding which initial states lead to which final steady states. For examples in nature, please see the answer to question 6.

2. In relation to the previous point, in figure 5 a, two situations corresponding to extreme K values are shown. The initial states are different, and considering that these initial states arise

stochastically (no effect of the K prior to this time point), how does the model operate on them differentially? Could it be that two different initial states give rise to different outcomes despite the same K value is operating? Conversely, given the same initial state, how do the K s operate, and what is the parameter space that leads to the bifurcation?

The reviewer is absolutely correct that two different initial states can give rise to different final outcomes for the same value of K (e.g. if K yields a bistable system). In particular, when K is chosen so that the system develops checkerboard patterns (e.g. $K = 0.1$), there are at least two stable states, one corresponding to a black square in the upper left-hand corner and the other to a white square in the upper left-hand corner. Which checkerboard is ultimately observed depends on the initial conditions and noisy influences during system evolution (we observed both orientations of checkerboards in our experiments with $K = 0.1$). Our theory also shows that in this case the homogeneous (non-patterning) state is unstable and thus cannot be observed. Similarly if the homogeneous state is stable, the checkerboard states will be unstable. In other words, if the system is capable of patterning, no set of initial conditions will cause it to reach a homogeneous outcome. Finally, the parameter space leading to bifurcation is given in Fig. 5b.

3. What were the experimental considerations in terms of rates of transcription/translation, thresholds thereof for application of the different illumination regimes (durations/time steps). It should be stated how and why the actual set ups were selected.

The reviewer brings up a very relevant point, namely, that the rate of cell response to input determines the timescale of system behavior. One of the strengths of our setup is that we do not need to explicitly model the network inside of a cell in order to implement cell-in-the-loop. Therefore, rather than explicitly estimate transcription/translation/translocation rates, we calculated a time constant for overall cell response (from light input to nuclear localization of fluorescent protein) and used this to choose the input adjustment time as described in the response to Reviewer 2, question 5. We have incorporated these details into Section 3.3, Supplementary Section S1.2.2, and the new Supplementary Section S.1.3.1 in the manuscript.

4. To extrapolate to more complex networks in vivo, the next logic step would be to couple the output to a signal that has a negative effect on the neighbouring cell, for instance that leads in a proof of principle to reduction of the output signal of a cell already taken as in the other state. In principle this would lead to a faster increase in the delta, and thus emergence of the pattern. What modules should incorporate the model to include this. What are the implications in terms of kinetics of the system? This is representative to what it happens in several systems occurring in nature.

We could certainly put expression of the reporter gene under the control of a repressor instead of an activator, such that increasing light intensity would decrease cell score. We would not need to incorporate any new modules into the theory; as before, we would simply need to measure the dose response and define the signaling relation. In this case, $h(\cdot)$ would be an increasing instead of a decreasing function to preserve lateral inhibition. The rate of pattern emergence depends on the kinetics of the promoter, so provided that the time constant for repression was smaller than the time constant for the activating promoter we used in our experiments (see response to Reviewer 2, question 5), it would be possible to achieve patterns more quickly than we did in our setup.

Alternatively, we could add an extra element into the network, using the light either to activate a repressor for reporter/target gene, or to repress an activator for reporter/target gene. Regardless, we would still not add any extra modules to the theory, and the procedure would remain the same as before: we would measure the dose response and define the signaling relation (as for the hypothetical example above, $h(\cdot)$ would now be increasing to preserve lateral inhibition). In this case the rate of pattern emergence would be limited by the slowest of the time constants among two transcription events, two translation events, and two nuclear translocation events (since transcription factor from the first promoter would need to relocalize to the nucleus to activate/repress the second promoter, and the reporter system we use also relies on translocation). We could then take a similar approach as before to measure the time constant from light input to fluorescent output.

5. Does the model consider cell growth and/or division? Would this be difficult to implement into it? [...] How robust are the systems? Is there any proxy of it? What happens when as in natural systems cells die, signals disappear, etc. Does the model incorporate this into account? This is particularly interesting in virtue of the limitations observed when the complexity of the system increases (e.g. 36 patches).

This is an excellent point. As is, our theory applies to systems with fixed numbers of interacting elements. Those elements can be cells, patches of cells, colonies of cells, etc. If those elements are individual cells, then the theory does not account for changes in cell number due to cell death, division, disappearance, etc., which affect not only the number of cells present but also which cells neighbor which. However, if the interacting elements comprise *multiple* cells, then the present theory may account for cell growth. For example, in our experiments, the interacting elements were patches of cells. By programming patch-patch signaling into the computer, we enforced that the number of patches remained constant. We kept the number of cells in a patch constant throughout an experiment by ignoring one daughter cell after each division event. Similarly, in previous work we applied this theory to a small system of bacterial colonies communicating via quorum sensing, and in the ODE models we explicitly included terms for dilution due to growth of bacteria within each colony (a colony here is mathematically equivalent to a patch in the present application; see Tei and Perkins et al. 2019). Even in systems where communication takes place between individual cells (rather than between patches or colonies), the current theory can still apply provided that the rate of gene expression/cell-cell signaling is faster than the rate of cell division. How much faster depends on factors such as how much divisions are synchronized and how large the system is—the larger the system, the less significantly any individual division or death affects the final outcome.

6. Overall, to increase the relevance for the general readership it would be interesting and useful to have clear examples of natural systems corresponding to the systems described in the article. In the same line, provide ideas, descriptions and discussion of putative engineered switches and cell synthetic signalling networks that could be developed in order to reconstruct the closed-loop system leading to bistability achieved by lateral inhibition in nature. A deeper and more complete discussion and perspectives should be included.

We appreciate the reviewer’s suggestions for increasing the relevance of our work for the general readership. We have incorporated new content into the Introduction and Discussion sections of the manuscript in the hopes of reaching a broader audience.

One of the motivating systems behind the theory was Notch/Delta contact-based cell-cell signaling. In fruit flies, where this system has been extensively studied, patterns shaped by mutual inhibition arise during wing vein patterning (Sprinzak et al. 2010/2011), as well as proneural stripe formation on the dorsal thorax and subsequent selection of isolated neural precursors from clusters of proneural cells (Heitzler and Simpson 1991, Corson et al. 2017). Vertebrates also rely on lateral inhibition for patterning in the inner ear (Bryant et al. 2002), intestine (Vooijs et al. 2011), and central nervous system (Lewis 1998, Pierfelice et al. 2011, Henrique and Schweisguth 2019).

On the synthetic side, lateral inhibition systems producing some kind of contrasting pattern have been implemented by the authors among colonies of *E. coli* communicating via quorum sensing (Tei and Perkins et al. 2019) and using exogenous Notch and Delta in CHO cells (Matsuda et al. 2015).

Regarding a synthetic lateral inhibition circuit that could be designed with the aid of a cell-in-the-loop system, we added the following paragraph to the Discussion section of the manuscript: “As an illustration, suppose one were to implement lateral inhibition with the basic transcriptional repressor circuit from (Matsuda et al. 2015), in which cleaved intracellular Notch domain activates a repressor for expression of Delta, the target molecule of Notch recognition domain. One might employ the synNotch toolbox for mammalian cells, which includes as design choices the recognition domain/target molecule pair and the transcription factor constituting the intracellular synNotch domain (Morsut and Roybal et al. 2016). One might begin with a cell-in-the-loop approach by engineering just the repressor under optogenetic control, and define computational variables for the concentrations of recognition domain, transcriptional activator (cleaved synNotch intracellular domain), and target molecule (plus intervening pathway components) in each cell. Using calculated activator concentration to modulate the light input, one could simulate different parameters for the remaining components to decide upon an activator. One could then engineer a circuit with repressor controlled by the chosen activator and place the activator under optogenetic control. With another round of cell-in-the-loop, one could further restrict the range of parameters for the combination of target molecule and recognition domain that would produce the desired effect *in vivo*, and use these numbers to guide construction of the final biochemical circuit in live cells.”

2 Reviewer 2

I commend the authors on a very interesting paper that tackles a crucial problem in the emerging field of pattern formation. Namely, how how a ”cell-in-the-loop” approach can be used to control pattern formation.

The manuscript reports the results of using a cell -in-the-loop approach for spatiotemporal control. Guided by theory, the authors engineered *S. cerevisiae* with a fluorescence reporter and emulated cell-cell signaling in in-silico via optogenetic control and showed that a virtual checkerboard pattern could emerge as a bifurcation parameter was varied. The results were in quantitative and qualitative agreement with the theory. This work stands as one of the first successful attempts to apply mathematical theory experimentally for multicellular patterning. The problem formulation is clearly stated and sufficient background and motivation is provided. The procedures and meth-

ods are clearly explained and sufficient details are provided for repeatability. The supplementary material complements the main text well and addressed a number of my questions.

Novelty and value to the field: This approach elegantly addresses many of the issues currently associated with spatiotemporal control. Primarily it bypasses having to construct explicit models of the intracellular and cell-cell dynamics that pure simulations require. This is advantageous, considering that the full structure of these models is usually unknown along with many of the parameters. Furthermore, simulations to these models may carry huge computational costs when individual cells are accounted for in a large population. By restricting cell-cell communication solely through user specified signals that are generated in-silico, this methods avoids issues that arise from crosstalk, stochastic effects, and burden on the cell from supporting large synthetic networks. Alleviating these unwanted interactions provides more reliable and replicable results. Through their implementation the authors had full access to the cell inputs and outputs, which is not something that is typically available in a pure biological experiment. This information provides insight that can be used in design or system identification. Even though only discussed briefly, this work provides insight on the role of stochastic effects in multicellular patterning and the limits where a deterministic model provides sufficient predictive power.

Overall, the manuscript is recommended for publication after the comments below are addressed.

We thank the reviewer for a thoughtful and supportive evaluation.

2.1 Major comments

1. Is there a particular reason the closed loop input to individual cells is in the form $h(\sum_i w_i)$ rather than $\sum_i h(w_i)$? The latter seems more biologically relevant.

The theory is actually general enough to handle arbitrary inputs of the form $g_1(\sum_i(g_2(w_i)))$ (see Arcak 2013, Ferreira and Arcak 2013). The terminology is a little unfortunate in that “input” in a physical system would refer to the molecules in one cell that diffuse to or bind molecules in another cell to initiate communication, whereas in a cell-in-the-loop system, the light is considered the input regardless of what aspects of a physical implementation might be simulated by the computer. Generally, portions of a genetic network downstream of reporter expression (determining what signal a cell transmits) should be encapsulated by g_2 while upstream elements (determining how a cell responds to received signal) should be encapsulated by g_1 .

In designing our particular system we thought of reporter expression as signal and chose $g_2(w_i) = w_i$ so that the input to the Hill function would be an average over signal from neighbors, based on a modeling paradigm for Notch/Delta in which the two inputs to a cell i are the averages over Notch and Delta in neighboring cells (see, e.g., Sprinzak et al. 2011 for modeling work in fruit flies, and Matsuda et al. 2015 for a synthetic lateral inhibition system implemented in mammalian cells). Although a “biologically accurate” model of contact-based lateral inhibition would include at a minimum two outputs per cell (Notch and Delta), for simplicity and clarity of exposition we used a single output, the nuclear localization score. As the reviewer points out, this may reduce the immediate biological relevance of the system we chose to implement, but we believe it is still sufficient as a general proof of concept. As a final note, systems with diffusion-based communication may also have inputs of the form $h(\sum_i w_i)$ where w is a diffusible molecule secreted by cells; see, e.g., Tei and Perkins et al. 2019.

2. Figure 3.c: Why are the error bars so small considering the large spread in the data?

The data do indeed have a large spread. The error bars represent estimated standard error of the mean for a given quantile (using the Bessel-corrected, i.e., unbiased estimation of population variance). There are 180 or 181 sample points per quantile, so even the maximum standard deviation of 0.275 that we observed for the bin with highest illumination intensity produces a standard error of just 0.0197 (as low as 0.00480 for the bin with lowest illumination intensity). We have clarified what the error bars are in the caption to Figure 3.

3. Figure 3.c: How is it possible to have a localization score greater than one?

This is a consequence of our definition of the nuclear localization score, since we averaged fluorescence independently over nucleus or cytoplasm or whole cell before subtraction and division. If we calculated the score via a “fluorescence proportion” across pixels (i.e., denominator were summed fluorescence over all pixels in cell and numerator were summed fluorescence over all pixels in nucleus or cytoplasm), then we would expect never to have a score above 1. (As an aside, with our definition of the nuclear localization score it is also theoretically possible to have a negative score, although we did not observe this to happen in our experiments.)

4. Figure 3.d and Figure S1.3: The variability between the 6-patch cells and the 4-patch cells seem quite similar. This is discussed briefly at the end of section 3.3 and it is claimed that patterning is not achieved due to the increased space of probabilities and not so much stochastic effects. Would it be possible to try the 36 patch test with 6 cells per patch just to rule this out?

This is a good point. We tried to run these experiments, but unfortunately we were unable to achieve a high enough density of cells in the field of view to fill out 36 patches with 6 cells/patch without having the cells grow out of a monolayer within 3 hours. However, some of our preliminary experiments addressed the same concern from a different angle using a system of 16 patches with 4 cells/patch (below, left), which achieved patterning within 3 hours. Even a system of 16 patches with 2 cells/patch (below, right) produced a checkerboard in < 3 hours, supporting our conclusion that stochastic effects are not the dominant cause of the failed patterning in the 36-patch case.

5. Results from Figure 3: Is it possible to estimate the time constant it takes for the cells to reach steady state? If so, how close is this to the 10 minutes at which the input was adjusted in the closed loop case? If larger, what are the pros and cons of increasing the input adjustment time?

To estimate the time constant, we ran an experiment with scores sampled every 2 minutes for cells switched from zero to the maximum intensity delivered during patterning experiments and back. We estimated the time constant as the time to reach half the maximum score, calculated from an average score over cells. For the rising edge we obtained a time constant of 10.9 min ($n = 63$ cells) and for the falling edge, 10.3 min ($n = 65$ cells). We have incorporated these details into Supplementary Section S1.2.2.

We chose the frequency of input adjustment to be on the same order of magnitude as the response time of the cells. Specifically, we decided to adjust inputs faster than the ~ 20 or 30 minutes for individual cells to reach steady state so that we could (a) ascertain when the *full system* (not just individual cells) had reached steady state by plotting the instantaneous input/output relation during the experiment (Figure 5), and (b) subject the theory to a more stringent test (i.e., make sure transient patch dynamics could still affect the dynamics of the full system without nullifying predictions made from single-cell steady-state behavior alone).

From a practical standpoint, the frequency of input adjustment is lower bounded by the amount of time it takes for image acquisition/the computer to automatically segment, track, and score cells (minimum ~ 2 min, somewhat longer the more cells there were in the field). To be absolutely safe we decided to adjust inputs no more frequently than every 5 min. We then figured that by 10 min, most cells would already have begun to respond measurably to an input, so we could avoid adjusting inputs more frequently than a change could be detected. We have incorporated this discussion into the main manuscript at the start of the second paragraph in Section 3.3.

In addition to the constraint placed by microscope/computer speed, adjustment time is also loosely upper bounded based on the practicality that cells will eventually grow out of a monolayer, which mandates that inputs be adjusted with enough frequency for a pattern to appear within 3 or 4 hours. Within the upper and lower bounds imposed by the setup, we expect that the adjustment time would primarily affect the speed of convergence to a pattern (slower inputs = slower convergence) as well as the frequency/magnitude of transient oscillations arising as a byproduct of discrete input adjustment (now Supplementary Section 1.3.2). We added a new supplementary section S1.3.1 to discuss these details.

6. The theory assumed all cells to have the same I/O properties to render the analysis of the 2-cell system sufficient for the full system (correct?). In experiments, this would correspond to each patch having identical I/O properties. If the dose response is repeated at the patch level, will the I/O responses be similar enough (6 cells per patch, 4 cells per patch)?

It is true that all cells were assumed to have the same I/O properties, and the reviewer is absolutely correct that this is relevant to the system. With the current constraints of our setup we can obtain responses for up to 4 doses for the same cell, which isn't enough sample points to fit a curve with 4 parameters such as our dose response curve. However, we can still simulate some patches by bootstrapping and visually compare linear interpolations across 4 doses with the empirical dose response curve. The results are shown below for 100 bootstrapped patches of 4 or 6 cells, all from a single dose experiment.

The patch-level I/O responses do exhibit some level of variation (higher for higher projected intensity), which probably manifests in the experiments as systematic variation in average patch intensity. This patch-patch variation, in turn, may be the most notable contributor to the experiment-to-experiment variation in exact contrast level and overall brightness observed in our patterning system (Fig. 5). Despite this variability, we were still able to qualitatively predict patterning and quantitatively predict average patch brightness, which we believe is strong evidence that deterministic theory can still adequately capture important features of full-system behavior even when stochastic influences are considerable. We have added this discussion to the new Supplementary Section S1.3.

7. Is it true that if $n = 1$ in $h(\cdot)$, then one does not expect patterning regardless of K ? If so, would it be worth showing that the experiment is consistent with the theory?

In our system it is indeed true that no patterning is expected when $n = 1$ regardless of K . Below on the left is the bifurcation plot for $n = 1$ (higher values of K were tested but not shown here):

The curve follows a similar shape to the homogeneous state in the $n = 2$ case (right).

In designing our system we had a choice of many bifurcation parameters—for example, we could have introduced a third if we multiplied $h(v)$ by a constant a —but for clarity we

wanted to consider just a single bifurcation parameter. Once we settled on Hill repression as a biologically meaningful function, we chose to vary K rather than n primarily because we believe it is more relevant to synthetic biologists. Specifically, K is an empirical concentration of repressor at which half maximum expression is achieved, whereas n represents cooperativity, which is to our understanding both more difficult to interpret and more difficult to control than K in an engineered genetic network (though some recent progress has been made, e.g., Bashor et al. 2019). There is also some empirical evidence that n is typically greater than 1 (e.g., Bashor et al. 2019, Figure 2b). Thus, we do not think varying K for fixed $n = 1$ would provide much more insight than has already been demonstrated.

2.2 Minor comments

- Page 8: In the first paragraph of section 3.3, a reference to Figure 4a is made even though it doesn't exist. Thanks for bringing this to our attention. We have fixed it to reference Fig. 4 alone.
- Figure 3.d: label the x-axis (cell-cell variation?) We've clarified the axis labels refer to cell-cell or patch-patch coefficient of variation (left) and relative variance (right).
- Figure 4: Perhaps is best to define the variables in the $h(v)$ equation We have added definitions in the figure caption. We also incorporated a minor clarification into the corresponding explanation in Section 5.6.2 to make clear that K was fixed within a given experiment.
- In 5.4 in is said that cells respond unpredictably to the first light dose and thus before any experiment they are illuminated for 10 minutes and then left in the dark for 20 minutes. Do we know why they respond this way? Also, have critical is the the time span of the initial illumination (the 10 minutes)? Can varying this change any of the results? At the moment we do not have a satisfactory explanation for why cells have this initial response. We do know that after it is done, cell response no longer depends on dose history (see Supplementary Section S1.2.4), from which we expect that longer initial illumination should not affect the experimental results.
- It is not clear how the periodic boundary conditions were enforced. Specifically during early time. The periodic boundary conditions are programmed into the computer through the choice of the matrix M , which instructs the computer as to which patches' scores should be averaged to determine the light input to any other given patch (i.e., it determines the "signaling neighbors" for a given patch). Since the matrix M does not itself contain information about the placement of patches on the virtual checkerboard, in theory we could make any set of patches signal any other set of patches regardless of whether they would be *spatial* neighbors when placed on a grid. To mimic a spatially interacting system, we chose to visualize patches that were signaling each other as spatial neighbors on the grid. The exception are the patches on the boundaries that only have two or three spatial neighbors. We arranged for these patches to signal patches on opposing boundaries, thereby establishing periodic boundary conditions. M is fixed throughout an experiment, so these conditions are enforced regardless of time.

To illustrate specifically for our system (where all patches have four signaling neighbors), if we index all patches, then the (i, j) th entry of M is equal to $\frac{1}{4}$ if patches i and j should signal

to each other, and 0 otherwise. So, for a 16-patch system with indexing as follows:

1 5 9 13
 2 6 10 14
 3 7 11 15
 4 8 12 16

we define

$$M = \begin{bmatrix} 0 & \frac{1}{4} & 0 & \frac{1}{4} & \frac{1}{4} & 0 & 0 & 0 & 0 & 0 & 0 & 0 & \frac{1}{4} & 0 & 0 & 0 \\ \frac{1}{4} & 0 & \frac{1}{4} & 0 & 0 & \frac{1}{4} & 0 & 0 & 0 & 0 & 0 & 0 & 0 & \frac{1}{4} & 0 & 0 \\ 0 & \frac{1}{4} & 0 & \frac{1}{4} & 0 & 0 & \frac{1}{4} & 0 & 0 & 0 & 0 & 0 & 0 & 0 & \frac{1}{4} & 0 \\ \frac{1}{4} & 0 & \frac{1}{4} & 0 & 0 & 0 & 0 & 0 & \frac{1}{4} & 0 & 0 & 0 & 0 & 0 & 0 & \frac{1}{4} \\ \frac{1}{4} & 0 & 0 & 0 & 0 & \frac{1}{4} & 0 & 0 & \frac{1}{4} & 0 & \frac{1}{4} & 0 & 0 & 0 & 0 & 0 \\ 0 & \frac{1}{4} & 0 & 0 & \frac{1}{4} & 0 & \frac{1}{4} & 0 & 0 & \frac{1}{4} & 0 & 0 & 0 & 0 & 0 & 0 \\ 0 & 0 & \frac{1}{4} & 0 & 0 & \frac{1}{4} & 0 & \frac{1}{4} & 0 & 0 & \frac{1}{4} & 0 & 0 & 0 & 0 & 0 \\ 0 & 0 & 0 & \frac{1}{4} & \frac{1}{4} & 0 & \frac{1}{4} & 0 & 0 & 0 & \frac{1}{4} & 0 & 0 & 0 & 0 & 0 \\ 0 & 0 & 0 & 0 & 0 & \frac{1}{4} & 0 & 0 & 0 & \frac{1}{4} & 0 & \frac{1}{4} & \frac{1}{4} & 0 & 0 & 0 \\ 0 & 0 & 0 & 0 & 0 & 0 & \frac{1}{4} & 0 & 0 & \frac{1}{4} & 0 & 0 & 0 & \frac{1}{4} & 0 & 0 \\ 0 & 0 & 0 & 0 & 0 & 0 & 0 & \frac{1}{4} & \frac{1}{4} & 0 & \frac{1}{4} & 0 & 0 & 0 & 0 & \frac{1}{4} \\ \frac{1}{4} & 0 & 0 & 0 & 0 & 0 & 0 & 0 & 0 & \frac{1}{4} & 0 & 0 & 0 & \frac{1}{4} & 0 & \frac{1}{4} \\ 0 & \frac{1}{4} & 0 & 0 & 0 & 0 & 0 & 0 & 0 & \frac{1}{4} & 0 & 0 & \frac{1}{4} & 0 & \frac{1}{4} & 0 \\ 0 & 0 & \frac{1}{4} & 0 & 0 & 0 & 0 & 0 & 0 & 0 & \frac{1}{4} & 0 & 0 & \frac{1}{4} & 0 & \frac{1}{4} \\ 0 & 0 & 0 & \frac{1}{4} & 0 & 0 & 0 & 0 & 0 & 0 & \frac{1}{4} & \frac{1}{4} & 0 & \frac{1}{4} & 0 & 0 \end{bmatrix},$$

where the red entries correspond to “wraparound” interconnections. These interconnections establish periodic boundary conditions because if we were to pick any starting index and move a distance of four signaling neighbors to the north, south, east, or west, we would return to that starting index. Here is a visualization of the resulting “periodized grid” with the wraparound interconnections highlighted in red:

- Did you test whether indeed you could illuminate a single cell without exciting any of its neighbors? Yes, we can illuminate single cells without exciting neighboring cells. Stochastically, sometimes cells will fail to respond, while “leakage” of light due to segmentation inaccuracies or cells being slightly out of focus might sometimes erroneously excite immediate neighbors. Here is a close-up example illustrating all these effects (red is a maximum projection fluorescence image, overlaid with the measured projected image in blue):

At the left center is an illuminated cell that is nicely activated with no neighbors responding. At the very top left is an illuminated cell that is not responding. At the very bottom right is a targeted cell with some possible excitation of its lower left (southwest) neighbor, probably due to leakage from the relatively large illuminated area. To reduce possible systematic effects from such leakage (and from any residual systematic variations in the illumination intensity across the plane), we randomized the placement of cells in patches such that no single patch would consist entirely of physically neighboring cells.

See also Rullan and Benzinger et al. (2018), Fig. 1, for an illustration of the targeting precision of the optogenetic system (with a different yeast strain).

Reviewers' Comments:

Reviewer #1:

Remarks to the Author:

The authors have addressed all issues/comments/requests raised. In the response letter all considerations and answers were included. They have in addition accordingly introduced into the main text and supplementary material what is useful to have modified and completed for the target broad readership. Therefore I support the publication of the article as in the revised version.

Reviewer #2:

Remarks to the Author:

The authors have adequately addressed all points of concern and confusions through their comments and the modified manuscript. I therefore support its publication as is.